# Exercise and Quality of Life (QoL) in Patients Undergoing Active Breast Cancer Treatment—Comparison of Three Modalities of a 24-Week Exercise Program—A Randomized Clinical Trial

**DOI:** 10.3390/healthcare12111107

**Published:** 2024-05-29

**Authors:** María-Pilar Suárez-Alcázar, M-Elena García-Roca, Eladio J. Collado-Boira, Paula Recacha-Ponce, Maria Dolores Temprado-Albalat, Pablo Baliño, María Muriach, Raquel Flores-Buils, Pablo Salas-Medina, Carlos Hernando, Ana Folch-Ayora

**Affiliations:** 1Faculty of Health Sciences, Jaime I University, 12071 Castellón de la Plana, Spain; malcazar@uji.es (M.-P.S.-A.); garciroc@uji.es (M.-E.G.-R.); recacha@uji.es (P.R.-P.); balino@uji.es (P.B.); muriach@uji.es (M.M.); flores@uji.es (R.F.-B.); psalas@uji.es (P.S.-M.); afolch@uji.es (A.F.-A.); 2Department of Medicine, Cardenal Herrera CEU University, 12006 Castellón de la Plana, Spain; maria.temprado@uch.ceu.es; 3Department of Education and Specific Didactics, Sport Service, Jaime I University, 12071 Castellón de la Plana, Spain; hernando@uji.es

**Keywords:** quality of life, breast cancer, chemotherapy, exercise program

## Abstract

Background: Exercise is an accepted intervention to improve the quality of life (QoL) of breast cancer patients. Exercise programs have been developed, and all have shown satisfactory results in improving the QoL. There is a lack of research comparing different prescription modalities. The aim of this study is to evaluate the effectiveness of physical exercise (in-person and home-based, compared to the exercise recommendation) on the QoL in breast cancer patients actively undergoing treatment. Methods: This is a randomized clinical trial with three groups (in-person: guided and supervised in-person exercise program; home-based exercise: guided and supervised exercise program with streaming monitoring both as a intervention groups; and recommendation: exercise recommendation as a control group). The QoL was measured using the EORTIC QLQ-C30 questionnaire. A baseline and 24-week analysis were investigated. Results: The total sample analyzed was n = 80. The QoL improved significantly at 24 weeks in the face-to-face and home-based exercise groups, but not in the control group. Exercise in all modalities improved fatigue, nausea, vomiting, appetite, and constipation. The QoL at 24 weeks depended on active chemotherapy, tumor type, and assigned exercise group (r^2^ = 0.503; *p* < 0.001). Conclusions: The QoL in breast cancer patients undergoing active treatment improved after a 24-week exercise program, especially in face-to-face and home-based exercise. Home-based exercise and streaming-based recommendation is a viable option for exercise recommendation.

## 1. Introduction

Breast cancer is the most common malignancy in women, thus accounting for 11.7% of all cancer diagnoses worldwide [1]. Its incidence is increasing due to increased life expectancy. Despite advances in early diagnosis and treatment, eradication involves therapies with numerous adverse effects during and after treatment. These include lymphedema, arthralgia, fatigue, osteoporosis, sleep disturbances, cardiotoxicity, peripheral neurotoxicity [2,3,4], anxiety, fear, and depression [4].

As a result, these patients often experience a poor QoL and overall well-being [5]. This concern has led to the exploration of new strategies to improve the lives of these patients, including both pharmacologic and nonpharmacologic interventions such as the recommendation of physical exercise [6,7,8].

Exercise is an accepted intervention to improve the QoL of cancer patients due to its benefits in cardiovascular and muscular health, as well as fatigue reduction [2,3,4]. Its practice is safe and feasible at all stages of the oncological process, including postoperative patients or patients utilizing other modalities of treatment [9], thus potentially reducing recurrence time, improving survival, and mitigating the side effects of cancer treatments [10]. Given these benefits, current guidelines recommend that physical activity be incorporated into the routine of cancer patients [9,11].

However, cancer patients often do not meet the minimum exercise guidelines [12], which recommend 150 min per week of moderate intensity aerobic exercise (heart rate 30–80%) or 75 min per week of vigorous exercise [13,14,15], combined with strength training 2–3 times per week [16]. Difficulty balancing daily routines, exercise, and medical appointments contributes to inadequate physical activity [17].

One of the main reasons for not exercising is the fear of causing harm or performing exercises that may be contraindicated in their health status [18]. Therefore, it is recommended that physical activity be performed in structured programs guided by oncology professionals [19], which provides women with a sense of security and increases adherence [20]. This aspect is crucial and challenging due to the scarcity of resources and trained personnel in the field of oncology. Oncology professionals will require instruction in delivering exercise counseling and/or facilitating suitable referrals to team members [21].

Alternatives such as remote session monitoring have been shown to be a good alternative to traditional brochures [22] to reach a larger population. Participation in face-to-face group exercise programs has shown very satisfactory results in the QoL of cancer patients [23]. Recently, new modalities such as home-based exercise have been incorporated [24]. While a considerable number of patients may have the capability to engage in independent exercise either at home or in community-centered environments, the establishment of a network comprising clinical or supervised exercise initiatives will be essential for others. This infrastructure would enable clinicians to make appropriate referrals, thereby directing patients to programs tailored to their specific requirements and capacities [21].

Since there are no studies comparing the three modalities of an exercise in-person group, home-based exercise, and recommendation, this study aims to determine the effectiveness of three modalities of exercise (in-person, home-based exercise, and recommendation) on the QoL of breast cancer patients actively undergoing treatment in the three dimensions of EORTIC QLQ-C30 overall health status, functional scores, and clinical symptoms scores [25].

## 2. Materials and Methods

### 2.1. Design

This is a clinical trial with randomized assignment to three groups (in-person: 24-week guided and supervised in-person exercise program; home-based exercise: 24-week guided and supervised exercise program with streaming monitoring, both as an interventions group; and recommendation: exercise recommendation by oncologist as a control group with baseline and 6-month analyses from the start of the study). The study period was October 2021 to July 2023 and conducted to respond to this investigation question: “what is the comparative effectiveness of three exercise modalities (in-person group, home-based exercise, and recommendation) on the QoL of breast cancer patients on active treatment?”

### 2.2. Population and Setting

Women diagnosed with breast cancer (stage I–IV) actively undergoing treatment (chemotherapy, radiotherapy, hormone therapy) constituted the study population. Exercise prescription should not be contraindicated by the oncologist, and participants must agree to participate in the study. Group assignment was based on group capacity (35 people per group) according to the group sequence: face-to-face, home-based exercise, referral. The estimated sample size was based on a 95% confidence level, a 5% margin of error, and a population of 105, thus resulting in a sample size of 80 patients. Recruitment was carried out at the Medical Oncology Service of the Provincial Hospital Consortium of Castellón. The study was conducted according to the Declaration of Helsinki, which was approved by the Human Research Ethics Committee of the Castellón Provincial Hospital Consortium (Protocol number 01/29/2020). Written informed consent was obtained from all participants prior to the study. The study, which currently lacks an identification code, has been registered on ClinicalTrials.gov (NTC06275321). This code will be provided during the article review process.

### 2.3. Intervention

The intervention consisted of a 24-week personal training program guided and supervised by a graduate in physical activity and sports—specialized in exercise and oncology and carried out in person or via home-based exercise—and a control group with only the recommendation given by the oncologist. The participants received a basal valuation after their inclusion in the program to evaluate their physical condition at the beginning of the program. The sessions consisted of a 10 min warm-up with joint mobility and balance exercises. This was followed by a 40 min main exercise session to improve upper and lower body strength and cardiorespiratory fitness, thus focusing on all major muscle groups and using body weight, resistance bands and/or free weights, exercise mats, and materials available at home (plastic bottles, shopping bags, etc.). This portion included a combined circuit of 8–12 functional exercises (e.g., squats, front and side lunges, sit-ups, calf raises, glute bridges, core, biceps curls, shoulder presses, punches, jumping jacks, static walking/jogging). The circuit consisted of 2 sets of 10–12 repetitions for the functional strength exercises and 30 s for the aerobic exercises. Volume was progressively increased by modifying the number of repetitions and sets and the complexity of the exercises. A minimum rest period of 90 s was established between exercises. For home-based exercise, the synchronously supervised home-based group participated in a home-based exercise program streamed and supervised by their oncology team for 6 months. Participants were asked to complete a 60 min combined resistance and aerobic exercise session two days per week for 6 months (24 weeks) as recommended in the latest guidelines [26]. The sessions were controlled, guided, and supervised by a cancer exercise specialist who encouraged and provided feedback to the participants while they could observe their performance, interact, or ask questions.

In the home-based exercise, Google Meet was used for the connection using the teacher’s and patient’s cameras and microphones. The exercise program was guided and supervised via live streaming, with the same training plan as in the face-to-face group. The recommendation of exercise prescription by the oncologist was our control group that did not receive the physical exercise intervention. Attendance was monitored for the face-to-face and home-based exercise, with a minimum attendance of 70% of sessions.

### 2.4. Variables

Baseline (before group assignment) and 6-month analysis were performed on sociodemographic variables: age (years), marital status (married or in a relationship, separated or divorced, single, widowed), motherhood (yes or no), cohabitation (living alone or not), education level (primary, secondary, university), employment status (employed, unemployed, retired) and income (<1000 EUR, 1000 to 2000 EUR, >2000 EUR); clinical variables: tumor type (luminal A, luminal B (HER2+), luminal B (HER2−), Enriched-her2, Basal-like), laterality (right breast, left breast, bilateral), tumor stage (I, II, III, IV), chemotherapy (yes or no), radiotherapy (yes or no), hormone therapy (yes or no). The European Organization for Research and Treatment of Cancer Quality of Life Questionnaire-C30 (EORTC QLQ-C30) stands as the predominant method employed globally for evaluating QoL among cancer patients [25].

The EORTIC QLQ-C30 questionnaire has 30 items covering five cancer dimensions: physical functioning (items 1–5), daily activities (items 6 and 7), social (item 27), emotional (items 21–24), and cognitive (items 20 and 25). It includes three symptom scales: fatigue (items 10, 12 and 18), pain (items 9 and 19), nausea and vomiting (items 14–15). It includes a global health scale (items 29–30) and individual items measuring disease and treatment symptoms: shortness of breath (item 8), insomnia (item 11), loss of appetite (item 13), constipation (item 16), diarrhea (item 17), and financial impact (item 28). It consists of a Likert-type response format referring to a one-week period. The EORTIC QLQ-C30 questionnaire was administered two times: at baseline and 24 weeks after the exercise intervention in the control and the experimental group.

### 2.5. Statistical Analysis

Statistical analysis was performed using IBM SPSS Statistics version 28 (IBM Corporation, Armonk, NY, USA). The normal distribution of variables was verified by the Kolmogorov–Smirnov test (*p* < 0.05). As the variables were not normally distributed, nonparametric statistical tests were applied. To describe the collected data, we used the mean and standard deviation for continuous variables and the frequency for categorical variables. The bivariate analysis (pre–post) of repeated measures was performed using the Wilcoxon test.

Multiple regression analysis was performed using the forward stepwise method. Only normally distributed variables were used as dependent variables. The parsimony principle was applied to the models obtained [27]. Given the limited simple size and the non-normal distribution of the independent variables, the residual errors of the resulting models were checked to ensure their normal distribution and thus the reliability of our regression models [27,28]. To determine the predictive value of the model, the Cohen criterion was applied to one-way ANOVA models. This criterion indicates that R2 values below 0.10 do not represent a relevant explanatory value, while R2 values between 0.10 and 0.25 indicate a dependence of the explanation of the variance of the analyzed variable on the identified factors, and with R2 values above 0.25, we can affirm that the explanatory model is very clinically relevant. Statistical significance is assumed at *p* value < 0.05.

To calculate the reliability of the data obtained, we calculated Cronbach’s alpha in the total result of the Global Health Scale and in the dimensions (physical functioning, daily activities, emotional, cognitive).

## 3. Results

### 3.1. Sociodemographic Description of the Sample

A total of N = 105 patients meeting the inclusion criteria were recruited, with study completion in the face-to-face group (n = 21; 26.3%), home-based exercise group (n = 31; 38.8%), and recommendation group (n = 28; 35%). The total sample analyzed was n = 80. See the flow chart for reasons for dropout by group (Figure 1).

The mean age was 48.6 years, including married marital status (n = 55; 68.8%), living with others (n = 72; 90%), secondary education (n = 35; 43.8%), employed (n = 60; 75.5%), and income between 1000 and 2000 EUR (n = 45; 56.3%), with no differences between groups according to allocation; see Table 1.

### 3.2. Description of the Clinical Status of the Sample

The description of the clinical status of the sample was obtained from the patients’ medical records facilitated by the oncologists. The most common tumor according to pathological anatomy was carcinoma NOS (n = 75; 93.75%) in the left breast (n = 43; 53.8%) and in stage II (n = 43; 53.8%). A total of 47 patients (58.8%) received chemotherapy during the study, 25 (31.2%) received hormonal therapy, and 8 (10%) received radiotherapy. The tumor stage was not considered as an exclusion criterion even for stage IV, the recruitment of the patients was carried out by the oncologists, and they did not refer to the physical exercise programs those who they considered would not be able to carry out the intervention.

No differences were observed between groups according to allocation; see Table 2.

### 3.3. QoL Results

As shown in Table 3, the global QoL improved significantly at 24 weeks in the face-to-face and home-based exercise groups, but not in the recommendation group (control group).

By dimension, the QoL worsened significantly (*p* value < 0.05) in activities of daily living (baseline = 43.40; 24 weeks = 37.67), social dimension (baseline = 50.30; 24 weeks = 44.84), and emotional dimension (baseline = 47.91; 24 weeks = 43.84).

For symptoms, significant improvements (*p* value < 0.05) were observed for fatigue (baseline = 51.31; 24 weeks = 46.49), nausea and vomiting (baseline = 33.10; 24 weeks = 26.86), loss of appetite (baseline = 35.94; 24 weeks = 26.92), and constipation (baseline = 45.63; 24 weeks = 33.33). In the home exercise group, there were significant improvements in nausea and vomiting (baseline = 37.68; 24 weeks = 27.42), appetite loss (baseline = 39.52; 24 weeks = 26.92), and constipation (baseline = 50.81; 24 weeks = 36.54). The recommendation group did not show statistically significant differences in any of the symptoms.

### 3.4. Reliability of the QLQ-C30

The reliability, based on the calculation of the global Cronbach’s alpha and by dimension, ranged from 0.762 to 0.906. See Table 4.

### 3.5. Regression Analysis Results

The QoL at 24 weeks depended on active chemotherapy, tumor type, and assigned exercise group, thus accounting for 50.3% of the variance (r2 = 0.503; *p* < 0.001).

Table 5 shows the explanatory model of variance (regression). 

## 4. Discussion

The QoL of breast cancer patients undergoing active treatment improved after a 24-week exercise program. These data support the importance of prescribing exercise during cancer treatment [2,3,4,9]. Physical activity is particularly effective in improving the QoL when delivered in person and virtually. These results are likely due to the development and supervision of these sessions by specialists in physical activity and cancer [19], which is consistent with the results of the Heiman study [3], which showed significant improvements in the QoL in patients who received guided and supervised exercise. Such supervision maximizes the benefits of exercise and helps women feel safe in their exercise routine [20].

In terms of dimensions, the QoL worsened significantly in the performance of activities of daily living, social dimension, and emotional dimension. No significant differences between the groups were observed in the analysis. Our results are not consistent with those of other researchers [22], who reported satisfactory results in improving the QoL in all dimensions, both in home-based exercise prescription modalities [29] and in face-to-face sessions [6]. Our results may be more related to changes in family and social dynamics after diagnosis than to the benefits of exercise prescription. In a recent investigation, García-Roc et al. [29] concluded that group-based physical exercise has the potential not only to instill self-esteem and address self-compassion but also to empower women to feel confident during their treatment and prevent cancer-related side effects. We cannot forget that most research has been conducted in breast cancer survivors who have completed their oncologic treatment and have experienced life changes after cancer [3,6,10,11,13,14,30].

Exercise in all modalities improved important aspects such as fatigue, nausea, vomiting, appetite, and constipation symptoms that often lead to treatment discontinuation or delay. The face-to-face group, and especially home-based exercise, showed greater benefits in reducing nausea, vomiting, appetite loss, and constipation. But, we must consider that nausea/vomiting and constipation would be mainly related to the time of chemotherapy and antiemetic therapy. A high number of the patients received chemotherapy during the intervention aspect to take into consideration when interpreting the results obtained. According to the meta analysis carried out by Chen et al. [25], there was no notable distinction observed in the manifestation of appetite loss and constipation symptoms between breast cancer patients and the control group comprising patients who refrained from exercising, but there were significant results in nausea and vomiting symptoms that there were significantly lighter than those in the control group who did not exercise.

These factors make home-based exercise and streaming-based programs resources to consider in exercise recommendation. Therefore, it is an alternative to offer guided and supervised exercise programs to a larger population, as confirmed by other researchers [30,31]. Furthermore, it should be noted that the average age of the patients is younger than expected, so the results may be different in an older population in the real world. One of the inclusion criteria was prescription by the oncologist; Due to this, the patient profile we receive does not correspond to the average of cancer patients. Currently, about 80% of patients with breast cancer are individuals aged > 50, while at the same time, more than 40% are those more than 65 years old [31].

The confidence level obtained from the EORTIC QLQ-C30 scale was 0.898, which yielded data higher than those obtained in its original validation with a confidence level of 0.846 [32]. Although its validation was carried out in patients with breast cancer 6 and 12 months after the end of their treatments and based on the results of Aune’s meta analysis [8], they establish that this questionnaire is not adequate to reflect the QoL in the short term, since it presents a large variability with respect to the effect size when evaluated during treatment. Recommended questionnaires for assessing the QoL during treatment are the FunctionalAssessment of Therapy (FACT) questionnaire, with its subscales of symptoms, physical well-being, and functional well-being, which have a Cronbach’s alpha values ranging from 0.55 to 0.76 [14]. In conclusion, the QoL is influenced by the type of treatment received during the trial, especially chemotherapy, which, along with surgery [33], has the greatest impact on QoL-related symptoms, as well as tumor type. Therefore, the explanatory model is associated with known QoL variables in breast cancer patients.

## 5. Conclusions

The QoL of breast cancer patients undergoing active treatment improved after a 24-week exercise program, especially in programs designed and supervised in-person and for home-based exercise. Home-based exercise using the streaming-based modality is a good option for exercise prescription. The face-to-face group, and especially home-based exercise, showed greater benefits in reducing nausea, vomiting, appetite loss, and constipation. Future investigations must describe in detail the type of program designed for this population, since, as we have seen, not all programs will have the same effect on the quality of life of these patients. Given the physical and psychological health benefits of regular physical exercise for this population, promoting physical activity in women diagnosed with and being treated for breast cancer must be an essential public health priority, and oncologists should be involved in prescribing physical exercise as part of cancer treatment.

## Figures and Tables

**Figure 1 healthcare-12-01107-f001:**
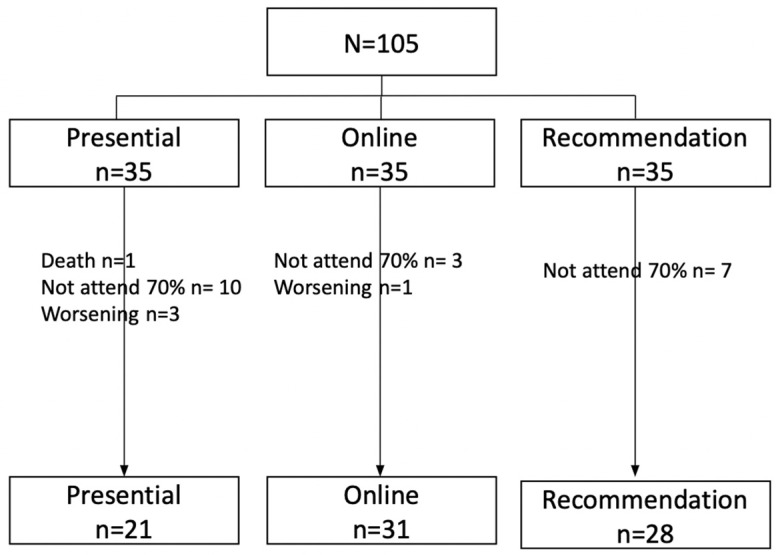
Flowchart of the process. Note: Flow chart with reasons for dropout per group.

**Table 1 healthcare-12-01107-t001:** Sociodemographic description of the sample.

	In Person n (%)	Online n (%)	Recommendation n (%)	Total n (%)
Age (mean ± sd)	46.1 ± 8.7	49.0 ± 8.9	50.1 ± 7.9	48.6 ± 8.6
Marital Status				
Married or in a relationship	14 (66.7)	24 (77.4)	17 (60.7)	55 (68.8)
Separated or divorced	2 (9.2)	4 (12.9)	5 (17.9)	11 (13.8)
Single	3 (14.3)	2 (6.5)	6 (21.4)	11 (13.8)
Widowed	2 (9.5)	1 (3.2)	0 (0.0)	3 (3.8)
Motherhood (yes)	15 (71.4)	25 (80.6)	22 (78.6)	62 (78.5)
Cohabitation				
No live alone	18 (85.7)	30 (96.8)	24 (85.7)	72 (90)
Live alone	3 (14.3)	1 (3.2)	4 (14.3)	8 (10)
Education Level				
Primary	3 (14.4)	6 (19.4)	5 (17.9)	14 (17.6)
Secondary	9 (42.8)	12 (38.7)	14 (50.0)	35 (43.8)
University	9 (42.8)	13 (41.9)	9 (32.1)	31 (38.8)
Employment Status				
Employed	18 (85.7)	23 (74.2)	19 (67.9)	60 (75.5)
Unemployed	2 (9.5)	5 (16.1)	8 (28.6)	15 (18.8)
Retired	1 (4.8)	3 (9.7)	1 (3.6)	5 (6.3)
Income				
<1000 EUR	5 (23.8)	13 (41.9)	8 (28.6)	26 (32.5)
1000–2000 EUR	12 (57.1)	15 (48.4)	18 (64.3)	45 (56.3)
>2000 EUR	4 (19.0)	3 (9.7)	2 (7.1)	9 (11.3)

List of socio-demographic variables collected from the participants in the study.

**Table 2 healthcare-12-01107-t002:** Description of the clinical status of the sample.

	In Person n (%)	Online n (%)	Recommendation n (%)	Total N (%)
Tumor Type				
Luminal A	7 (33.3)	12 (38.7)	10 (35.7)	29 (36.3)
Luminal B (her2+)	3 (14.3)	4 (12.9)	5 (17.8)	12 (15.0)
Luminal B (her2−)	9 (42.8)	11 (35.4)	10 (35.7)	30 (37.5)
Enriched-her2	1 (4.8)	2 (6.5)	2 (7.2)	5 (6.2)
Basal-like	1 (4.8)	2 (6.5)	1 (3.6)	4 (5.0)
Laterality				
Right breast	10 (47.6)	10 (32.3)	10 (35.7)	30 (37.5)
Left breast	9 (42.9)	17 (54.8)	17 (60.79	43 (53.8)
Bilateral	2 (9.5)	4 (12.9)	1 (3.6)	7 (8.7)
Tumor stage				
I	5 (23.8)	9 (29.0)	13 (46.4)	27 (33.8)
II	11 (52.4)	20 (64.5)	12 (42.9)	43 (53.8)
III	3 (14.3)	1 (3.2)	1 (3.6)	5 (6.2)
IV	2 (9.5)	1 (3.2)	2 (7.1)	5 (6.2)
Treatment during the study				
Chemotherapy	14 (66.7)	16 (51.6)	17 (60.7)	47 (58.8)
Radiotherapy	2 (9.5)	4 (13.0)	2 (7.1)	8 (10.0)
Hormonotherapy	5 (23.8)	11 (35.5)	9 (32.2)	25 (31.2)

Note: Clinical variables assessed in the selected sample.

**Table 3 healthcare-12-01107-t003:** QoL Results.

	Groups	Basal	24 Weeks	*p* Value/d Cohen
**Global Health**				
	Total	70.63 (±16.96)	77.25 (±14.29)	<0.001/0.53
	In person	75.14 (±13.26)	81.71 (±13.67)	0.028/0.55
	Online	69.35 (±16.40)	77.27 (±13.83)	0.005/0.60
	Recommendation	68.64 (±19.77)	72.95 (±14.72)	0.167
**Dimensions**				
Physical Functioning	Total	34.01 (±10.44)	34.20 (±11.93)	0.959
	In person	30.24 (±6.01)	30.48 (±4.71)	0.958
	Online	36.77 (±12.01)	33.08 (±10.87)	0.203
	Recommendation	39.09 (±10.59)	33.75 (±16.08)	0.179
Daily Activities	Total	43.40 (±19.92)	37.67 (±16.28)	0.071
	In person	35.76 (±14.41)	33.19 (±12.78)	<0.001/0.37
	Online	47.32 (±19.87)	37.54 (±15.81)	0.014/0.47
	Recommendation	44.79 (±22.43)	42.09 (±19.10)	0.198
Social	Total	50.30 (±21.93)	44.84 (±22.94)	0.018/0.35
	In person	50.76 (±22.85)	45.95 (±19.87)	0.324
	Online	53.74 (±22.41)	47.69 (±23.98)	0.195
	Recommendation	46.14 (±20.75)	40.41 (±24.72)	0.139
Emotional	Total	47.91 (±17.95)	43.84 (±18.26)	0.014/0.26
	In person	53.43 (±16.54)	51.90 (±19.33)	0.646
	Online	49.29 (±21.97)	39.73 (±15.41)	0.019/0.53
	Recommendation	42.25 (±13.68)	41.01 (±18.62)	0.549
cognitive	Total	43.28 (±17.60)	43.09 (±18.67)	0.703
	In person	41.76 (±15.43)	41.06 (±16.54)	0.717
	Online	44.13 (±15.82)	42.04 (±17.33)	0.843
	Recommendation	45.59 (±22.37)	43.89 (±21.15)	0.591
**Symptoms**				
Fatigue	Total	51.31 (±18.56)	46.49 (±14.61)	0.007/0.31
	In person	47.14 (±14.27)	45.19 (±9.34)	0.422
	Online	54.74 (±20.22)	45.54 (±14.07)	0.061
	Recommendation	50.64 (±19.37)	48.86 (±19.07)	0.259
Pain	Total	46.69 (±19.20)	46.75 (±16.72)	0.456
	In person	41.24 (±13.86)	43.67 (±14.62)	0.458
	Online	52.52 (±22.44)	47.35 (±13.81)	0.633
	Recommendation	44.32 (±17.56)	49.01 (±31.42)	0.204
Nausea and Vomiting	Total	33.10 (±12.74)	26.86 (±6.27)	<0.001/0.52
	In person	30.43 (±9.39)	26.24 (±3.91)	0.633
	Online	37.68 (±15.25)	27.42 (±8.75)	0.009/0.71
	Recommendation	30.04 (±10.58)	26.77 (±4.56)	0.084
Shortness of breath	Total	30.94 (±12.73)	31.01 (±13.27)	0.674
	In person	29.76 (±10.06)	29.29 (±10.52)	0.928
	Online	33.06 (±16.31)	28.85 (±9.19)	0.317
	Recommendation	29.46 (±9.75)	35.23 (±18.35)	0.157
Insomnia	Total	53.75 (±25.50)	56.88 (±24.96)	0.249
	In person	59.52 (±27.92)	63.10 (±23.21)	0.509
	Online	54.84 (±26.94)	53.85 (±24.18)	0.593
	Recommendation	48.21 (±21.44)	54.55 (±27.43)	0.053
Loss of appetite	Total	35.94 (±18.16)	28.99 (±11.03)	0.005/0.32
	In person	33.33 (±12.07)	29.76 (±10.06)	0.317
	Online	39.52 (±21.18)	26.92 (±6.79)	0.013/0.55
	Recommendation	33.93 (±18.27)	30.68 (±15.29)	0.589
Constipation	Total	45.63 (±26.91)	33.33 (±16.42)	<0.000/0.47
	In person	40.48 (±24.33)	29.76 (±12.79)	0.002/0.45
	Online	50.81 (±29.21)	36.54 (±20.28)	0.002/0.56
	Recommendation	43.75 (±26.02)	32.95 (±14.19)	0.047
Diarrhea	Total	31.25 (±15.15)	28.62 (±9.85)	0.361
	In person	30.95 (±13.47)	28.57 (±8.96)	0.414
	Online	29.03 (±9.35)	29.81 (±12.28)	0.414
	Recommendation	33.93 (±20.65)	27.27 (±7.35)	0.157
Financial impact	Total	40.31 (±24.03)	41.30 (±24.94)	0.859
	In person	34.52 (±18.50)	42.86 (±27.54)	0.107
	Online	46.77 (±27.94)	44.23 (±25.79)	0.527
	Recommendation	37.50 (±22.04)	36.36 (±21.44)	0.334

Note: *p* < 0.05 between intragroup values between scores at baseline and at 24 weeks.

**Table 4 healthcare-12-01107-t004:** Reliability scale.

	Cronbach’s Alpha
Global Health	0.898
Physical Functioning	0.762
Daily Activities	0.829
Social	0.906
Emotional	0.886
Cognitive	0.877

**Table 5 healthcare-12-01107-t005:** Explanatory model of variance (regression).

Model	R2 Adjusted	Standard Error	B	F (*p*)
Dependent Variable: global health scale at 24 weeks into the program Covariates: Tumor type. Chemotherapy and Type of physical exercise program	0.503	9.710	59.215	14.515 (<0.001)

## Data Availability

For ethical reasons related to the preservation of patient identity, the data presented in this study are available upon request to the corresponding author.

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
