# Peer review of "Exercise and Quality of Life (QoL) in Patients Undergoing Active Breast Cancer Treatment—Comparison of Three Modalities of a 24-Week Exercise Program—A Randomized Clinical Trial"

_healthcare, 2024, doi:10.3390/healthcare12111107_

Round 1
Reviewer 1 Report
Comments and Suggestions for Authors
First of all, thank you very much for allowing me to review this very interesting masnucrito. Here are some recommendations for the authors to consider.
Introduction
The first 2 lines of the introduction provide statistical data, please cite the source of the information.
In the abstract they state the acronym QoL, however, on line 56 the term appears as "quality of life", please use the acronym throughout the manuscript.
I do not understand the 3 lines referring to the measurement instrument in the introduction, this information is more appropriate in the Methods section.
Materials and Methods
The subsections of the section are not in the proper journal format.
However, the rest of the section is appropriate.
Results
The tables are not in the correct format.
Also, I do not understand why they do not provide all the p-values in Table 3 and identify those that are significant. Also, decide whether the values will appear to two or three decimal places and apply it throughout the paper.
Similarly, I find it intriguing that the authors do not include the standardized Beta coefficients in the predictive regression model, so that the reader is aware of the extent to which each independent variable influences the model.
Discussion
The year of publication of the reference appears on line 272.
The Discussion is appropriate, however, it seems to me to be too few for the number of variables analyzed in the study.
First of all, thank you very much for allowing me to review this very interesting masnucrito. Here are some recommendations for the authors to consider.
Introduction
The first 2 lines of the introduction provide statistical data, please cite the source of the information.
In the abstract they state the acronym QoL, however, on line 56 the term appears as "quality of life", please use the acronym throughout the manuscript.
I do not understand the 3 lines referring to the measurement instrument in the introduction, this information is more appropriate in the Methods section.
Materials and Methods
The subsections of the section are not in the proper journal format.
However, the rest of the section is appropriate.
Results
The tables are not in the correct format.
Also, I do not understand why they do not provide all the p-values in Table 3 and identify those that are significant. Also, decide whether the values will appear to two or three decimal places and apply it throughout the paper.
Similarly, I find it intriguing that the authors do not include the standardized Beta coefficients in the predictive regression model, so that the reader is aware of the extent to which each independent variable influences the model.
Discussion
The year of publication of the reference appears on line 272.
The Discussion is appropriate, however, it seems to me to be too few for the number of variables analyzed in the study.
Author Response
Thank you very much for taking the time to review this manuscript. Please find the detailed responses below and the corresponding corrections highlighted in the re-submitted files.
Introduction
- We have included the source of the statistical data provided in the first line of the introduction. We apologize the mistake.
- We have correct the term "Quality of life" with its acronym all over the document.
- We have moved the lines on the measuring instrument to the method section. We considered is the most appropriate too.
Materials and method
- We have change the subsections in the method section to fit them correctly.
Results
- Tables have been modified and all of the p-values with the same number of decimals have included. It has also been included the standardized Beta Coefficients in the predictive prediction regression model.
Discusión
- We have eliminated the date of the publication.
- We appreciated the comment; it is true that the number of variables is high, but we consider that those that have shown significant differences have been fully incorporated into the discussion.
Reviewer 2 Report
Comments and Suggestions for Authors
This is a very strong article that is clear and well-written. It makes an important contribution to the literature adding evidence that exercise is helpful to women undergoing breast cancer treatment.
The literature review is good with up-to-date references.
The Materials and Methods section is clear. A sample of 80 is adequate for this work. What makes the Intervention section especially strong is that the authors describe the actual exercises.
I am not a research methodologist/statistician and I hope another reviewer can be helpful here.
Graph 1 is a good display of the how the sample was determined. Should this be labeled Figure 1? Table 1 nicely depicts the sample demographics. Table 2 describes the severity of breast cancer.
We learn that the in-person exercise and online-based based exercise are very helpful, while the simple oncologist recommendation for exercise is not. Table 3, the quality of life results is excellent. It is very useful to learn the exact symptoms patients experience that do improve with exercise.
It is very important that the authors note the dimensions in which quality of life worsened--activities of daily living, social dimension, and emotional dimension. I'd like to see the authors address this a bit more. Certainly women may be depressed and feel isolated, friends and family may have pulled back to deal with their own feelings, and there may be confusion as to how to support the women who have always supported them. I'd like the authors to share their thoughts a bit more, perhaps after line #262.
The Conclusions section is a bit weak. After all their good work, I want to encourage the authors to share more detail about where to go from here. What are their thoughts about how to get oncology offices to move beyond mere recommendation for exercise and make specific arrangements for exercise programs? What are the authors' suggestions for other researchers and their own future work? Comparing specific kinds of exercise such as a general aerobic and strength exercise program with yoga, known to have strong physical and mental benefits?
Overall, very strong work!
Author Response
Thank you very much for taking the time to review this manuscript. Please find the detailed responses below and the corresponding corrections highlighted in the re-submitted files.
- The nomenclature indicate for graph 1 have been changed to Figure 1 which we considerer more appropriate. Thank you so much for the contribution.
- We hope with the contributions made in the discussion in relation to other research in with several researchers components of this manuscript participated contribute a little bit more to clarify how to support this women.
- We have expanded the conclusion a bit further to include where new research should be directed and the importante of involving of oncologists in prescribing exercise as a part of cancer treatment.
Reviewer 3 Report
Comments and Suggestions for Authors
Much of the discussion included results. I would like to see more of a synthesis and then a comparison to the literature to support or refute your study.
The conclusion should be longer. It would be helpful to include recommendations, implications, and suggestions for future research.
This manuscript does have significance for a specific reading audience; generalists may find it applicable.
specific comments such as:
1. While this is identified as a quasi-experimental study with random assignment (line 97-98) to three groups, variables were identified, but no specific research questions were identified. It is recommended the authors determine the research questions and add them to this manuscript in the methods section. Quasi-experimental studies are NON-randomized.
2. Treatment modalities, including exercises, for breast cancer patients are vital for the women to regain the use of the affected muscle groups. This part of the introduction was not clearly spelled out. Were these participants post-operative patients for whom exercise is strategic, or patients utilizing other modalities (chemotherapy, radiotherapy, hormone therapy).
2a. The authors do articulate the lack of studies concerning exercise regimens that compare the three groups identified by the researchers: in-person groups, home-based groups, and recommendations (control group). Dividing the participants into these groups, the authors utilized a quality-of-life instrument to determine the results for cancer patients and exercise.
3. This manuscript adds quantitative data to the quality-of-life issues breast cancer patients face as they migrate through the stages of emotional windstorm in dealing with the diagnosis and life after. Combining exercise in social settings allows those patients to mingle and discuss how they are managing, much like a therapy session. As the authors have identified this area as lacking in journals, this study adds to the body of knowledge currently available.
4. The authors identify the design of this study as quasi-experimental with randomized assignment to three groups. Various sources describe quasi-experimental design as: non-random assignment to groups, can be conducted to evaluate the effectiveness of a treatment or intervention, may have a control group. As no specific experimental group was identified, lines 162-164 are confusing for the reading audience. A succinct outline and definition of a quasi-study would help in the logical flow of the narrative.
4a. By outlining the format of the narrative presentation, the identification of the type of this quasi-study would further provide clarity. It is inferred that this could be a non-equivalent study, however, without a definite indication, it could also be another form of quasi-study.
5. As the conclusions are currently listed, the authors need to expand on limitations, recommendations, and comparison of their study to the ones they have identified, continuing the narrative on the gap and if their study added anything to the literature.
5a. As there were no research questions identified for this manuscript, the study only addressed the exercise domains and the study population of breast cancer patients and their inclusion in the three groups designated by the researchers.
One exception of note would be the rationale and justification for the randomization of the target population, which is not a means of assignment for participants in quasi-experimental studies. It appears as if the authors have intermingled experimental and quasi-experimental in some areas of the manuscript (lines 242-243).
6. I didn't see any references concerning radiotherapy (line 105). Most of the references involved RCT or literature reviews. Most are within the current five-year time frame.
7. Tables lacking full description of results. P-values missing either as a note for the tables or annotated in the header, with an indication of the significant results, and in the narrative, why that was important.
Comments on the Quality of English LanguagePlease review numerical annotations for 10 and under and the beginning of a sentence.
Author Response
Thank you very much for taking the time to review this manuscript. Please find the detailed responses below and the corresponding corrections highlighted in the re-submitted file.
First of all, thanks a lot for your comment. It is indeed and error on our part in classifying the study. Patients were indeed randomized at enrollment and it should be considered a randomized clinical trial as started in the US Clinical Trials registry of which this trial is a part. The study is prospective, the patients were randomized according to the time of enrollment, an there is a control group. We have corrected these aspects in the methodology and in the tittle of the article. We hope that this clarification will be accepted, we apologize for our error and we hope that this will remove the doubts that have arisen.
- We have included in the methodology section the research question to which we seek to answer.
- We clarify that as we have indicated “exercise is an accepted intervention to improve the quality of life of cancer patients. Its practice is feasible at all stages of the oncological process including post-operative patients or patients utilizing other modalities of treatment (chemotherapy, radiotherapy or hormone therapy).
- Tables have been modified and all of the p-values with the same number of decimals have included. It has also been included the standardized Beta Coefficients in the predictive prediction regression model.
Round 2
Reviewer 1 Report
Comments and Suggestions for Authors
The authors have largely corrected the issues pointed out during the first revision, providing the manuscript with quality.
Inhotabuena to the authors, the manuscript is currently publishable.